# Effects of Transcranial Direct Current Stimulation Combined with Treadmill Training on Kinematics and Spatiotemporal Gait Variables in Stroke Survivors: A Randomized, Triple-Blind, Sham-Controlled Study

**DOI:** 10.3390/brainsci13010011

**Published:** 2022-12-21

**Authors:** Arislander Jonathan Lopes Dumont, Veronica Cimolin, Rodolfo Borges Parreira, Danilo Armbrust, Daniela Rosana Pedro Fonseca, Adriano Luís Fonseca, Lorraine Cordeiro, Renata Calhes Franco, Natália Almeida Carvalho Duarte, Manuela Galli, Cláudia Santos Oliveira

**Affiliations:** 1Movement Analysis Lab, University of Sorocaba, Rodovia Raposo Tavares km 92.5, Sorocaba 18023-000, Brazil; 2Department of Electronics, Information and Bioengineering, Politecnico di Milano, Piazza Leonardo da Vinci 32, 20133 Milan, Italy; 3Istituto Auxologico Italiano, IRCCS, S. Giuseppe Hospital, 28824 Piancavallo, Italy; 4Health Sciences Program, Santa Casa School of Medical Sciences of São Paulo, St. Jaguaribe 155, São Paulo 01224-001, Brazil; 5Human Movement and Rehabilitation, Post-Graduate Program Medical School, Evangelic University of Goiás—UniEVANGÉLICA, Anápolis 75083-515, Brazil; 6Departamento de Fisioterapia, University Center of Americas, Campus Consolação, Street Augusta 1508, São Paulo 01304-001, Brazil

**Keywords:** stroke, transcranial direct current stimulation, gait

## Abstract

The present study assessed the effects of anodal transcranial direct current stimulation (tDCS) combined with treadmill training on spatiotemporal and kinematic variables in stroke survivors using gait speed as the primary outcome. A randomized, sham-controlled, triple-blind, study was conducted involving 28 patients with hemiparesis allocated to two groups. The experimental group was submitted to treadmill training combined with anodal tDCS over the primary motor cortex (M1) of the damaged hemisphere. The control group was submitted to treadmill training combined with sham tDCS. Stimulation was administered (2 mA, 20 min) five times a week for two weeks during treadmill training. No significant differences (*p* > 0.05) in spatiotemporal variables were found in the intra-group and inter-group analyses. However, the experimental group demonstrated improvements in kinematic variables of the knee and ankle (*p* < 0.05) and these results were maintained one month after the end of the intervention. The inter-group analysis revealed significant differences (*p* < 0.05) with regard to the pelvis, hip and knee. Anodal tDCS over M1 of the damaged hemisphere combined with treadmill training did not affect spatiotemporal variables, but promoted improvements in kinematic variables of the pelvis, hip, knee and ankle and results were maintained one month after treatment.

## 1. Introduction

Compromised gait due to hemiplegia or hemiparesis resulting from the total or partial loss of motor function is one of the most evident manifestations of a stroke [1]. Approximately 80% of stroke survivors have hemiparesis, with consequent muscle weakness, spasticity and stiffness [2] in the affected limb, leading to an abnormal gait pattern, which alters the spatiotemporal and kinematic characteristics of gait [3]. Hemiparetic gait is characterized by changes in the swing phase (prolonged on the affected side) and stance phase (diminished on the affected side) due mainly to insufficient plantar dorsiflexion, knee flexion and hip extension [4]. The poor kinematic performance leads to changes in spatiotemporal gait variables (velocity, step width, stance, double support and step length) [5]. The main changes are a reduction in walking speed and a longer double support phase [3].

The literature describes different techniques for gait rehabilitation, such as treadmill training [6], which is based on theories of neuroplasticity, motor learning, memory and neuromotor activation [7]. According to Werner et al. [8] and Ribeiro et al. [9], treadmill training offers the repetition of a specific task, promoting long-term improvements in spatiotemporal and kinematic gait variables.

Innovative methods have emerged to enhance the effects of motor training and minimize the costs and duration of the motor rehabilitation process. Transcranial direct current stimulation (tDCS) is easily used during motor training [10]. This noninvasive method is employed to stimulate the cerebral cortex using a low-intensity electrical current (1 to 2 mA) through electrodes positioned on specific parts of the scalp. The anode facilitates and the cathode inhibits cortical excitability. Thus, tDCS can increase local synaptic efficacy [11], changing the maladaptive plasticity of the damaged cortex following a stroke [12]. A large number of studies in the literature involving the use of tDCS on stroke survivors focus on upper limb function, but some researchers have begun to investigate the effects of this therapeutic modality on lower limb function [10,12,13].

No studies were found in the literature addressing the effects of tDCS combined with treadmill training on spatiotemporal and kinematic gait variables in this population. The hypothesis tested in the present investigation is that anodal tDCS over the damaged motor cortex can enhance the effects of treadmill training. Therefore, the aim of this study was to investigate the effects a single session as well as 10 sessions of anodal tDCS combined with treadmill training on spatiotemporal and kinematic gait variables in stroke survivors and determine whether these effects are maintained one month after the end of the 10-session intervention.

## 2. Materials and Methods

### 2.1. Study Design

A randomized, sham-controlled, triple-blind (assessor, therapists and patients) study was conducted in accordance with the principles contained in the Declaration of Helsinki and guidelines for research involving human subjects. This project received approval from the Human Research Ethics Committee under process number 575519 and is registered with the Brazilian Clinical Trials Registry under process number RBR-7n2d2f. All volunteers received clarifications regarding the objectives and procedures and signed a statement of informed consent agreeing to participate in the study.

### 2.2. Participants

Patients with a medical diagnosis of chronic stroke six months to five years [14] after a single ischemic or hemorrhagic event and who met the eligibility criteria were included. The inclusion criteria were either sex, age 45–60 years, hemiparesis from a single stroke event with crural predominance occurring six months to five years earlier and conventional standardized physical therapy at university clinics. Other inclusion criteria were the ability to walk barefoot with or without a gait-assistance device, controlled and clinically stable comorbid diseases, no metal implants in the head, the capacity to read information charts and understand people and the capacity to sign a statement of informed consent. Individuals with deformities of the lower limbs, those having undergone treatment with botulinum toxin and/or a neurolytic block in the previous six months, those with a history of osteoarticular disorders, those with any health condition besides stroke that would affect gait performance, those with cognitive impairment that would affect the performance on the tests, those who had undergone surgery and those who did not meet the inclusion criteria were excluded from the study.

### 2.3. Blinding

An assessor who was blinded to treatment allocation conducted all clinical assessments. Neither the therapists nor participants were aware of whether a sham or active treatment was being administered. The same tDCS device was used for both groups. To ensure blinding, the tDCS device emitted the same sounds and displayed the same information in both active and sham modes. The researcher who evaluated the outcomes and the researcher who performed the data analysis were also unaware of the group to which the participants were allocated. Only the researcher in charge of the randomization process and programming of the tDCS device had the identifying code to determine which treatment should be administered. This researcher was instructed not to disclose whether the patient would receive active treatment or sham to any of the participants or other researchers until the end of the study.

### 2.4. Randomization

The intervention consisted of 10 sessions (five sessions over a two-week period). The experimental group (EG) received active tDCS and the control group (CG) received sham stimulation administered for 20 min during treadmill training. The treatment order was randomized using codes generated through the random.org website. Randomization was balanced (1:1) to ensure equal distribution between groups. In the first session, each patient was allocated according to the randomization code (A or B), which determined allocation to the treatment group: A (active tDCS combined with treadmill training) or B (sham tDCS combined with treadmill training). Allocation concealment was achieved using sequentially numbered, sealed, opaque envelopes.

### 2.5. Outcome Measurements

The primary outcome was self-selected walking speed. The secondary outcomes were other spatiotemporal variables and kinematic variables of gait. All participants were evaluated one week prior to the intervention (Evaluation 1—baseline), immediately after one session (Evaluation 2), after 10 sessions (Evaluation 3) and at the one-month follow up (Evaluation 4). A member of the research team who did not interact with the participants during the interventions or evaluations exported the data to spreadsheets and sent the data to the statistician.

An identification chart was filled out containing the following patient data: name, telephone number, address, age, sex, height and weight.

Gait analysis was performed using the SMART-D 140^®^ program (BTS Engineering, Milan, Italy) with a sampling rate of 100 Hz, eight cameras sensitive to the infrared spectrum and the SMART-D^®^ (BTS Engineering, Milan, Italy) with 32 analog channels. All participants wore swimsuits to facilitate the placement of the reflective markers. After the anthropometric measurements (height, weight, lower limb length, distance between the femoral condyles or diameter of the knee, distance between the malleoli or diameter of the ankle, distance between the anterior superior iliac spines or thickness of the pelvis and vertical distance on the sagittal plane in the supine position between the anterior superior iliac spine and great trochanter), passive markers were placed at specific reference points directly on the skin to evaluate the kinematics of each segment of the body, as described by Davis et al. [15] After placement of the reflective markers, the participants were instructed to walk along a 10-m track at comfortable pace. At least six trials were performed (two sets of three). For each participant, three out of the six trials that were consistent in terms of gait pattern were considered for analysis. All data were exported in .txt format to electronic spreadsheets and tabulated using Microsoft Office Excel^®^ 2013 (Redmon, IL, USA).

The following were the spatiotemporal variables: velocity (m/s) (mean velocity of progression), step length (m, longitudinal distance between the point of initial contact of one foot and the point of initial contact of the contralateral foot), step width (m, distance between the rear end of the right and left heel centerlines along the mediolateral axis), stance phase (% gait cycle that begins with initial contact and ends with toe-off of the same limb) and double support (s, period of time when both feet are in contact with the ground).

The following were the kinematic variables (degrees0: Pelvis, PT-IC = angle of pelvic tilt at initial contact; PT-MAX = maximum angle of pelvic tilt; PT-MIN = minimum angle of pelvic tilt; PT-ROM = range of motion of pelvic tilt; PO-MAX = maximum angle of pelvic obliquity; PO-MIN = minimum angle of pelvic obliquity; PO-ROM = range of motion of pelvic obliquity; PR–MAX = maximum angle of pelvic rotation; PR-MIN = minimum angle of pelvic rotation; PR-ROM = range of motion of pelvic rotation. Hip, HIC = angle of hip flexion at initial contact; HMSt-MAX = maximum angle of hip flexion/extension in stance phase; HMSt-MIN = minimum angle of hip flexion/extension in stance phase; HMSt-ROM = range of motion of hip flexion/extension in stance phase; HAA-MAX = maximum angle of hip abduction/adduction; HAA-MIN = minimum angle of hip abduction/adduction; HAA-ROM = range of motion of hip abduction/adduction; HROT-IC = range of motion of hip rotation at initial contact; HROT-MEAN = mean value of hip rotation. Range of motion was computed as the difference between the maximum and minimum values of the specific plot. Knee, KIC = angle of knee flexion at initial contact; KMSW = maximum angle of knee flexion in swing phase; KmST = minimum angle of knee flexion in stance phase; K-ROM = range of motion of knee on sagittal plane. Range of motion was computed as the difference between the maximum and minimum (KmSt index) values of the plot. Ankle, AIC = angle of ankle dorsiflexion/plantar flexion at initial contact; AMSt = maximum angle of ankle dorsiflexion in stance phase; AmSt = minimum angle of ankle plantar flexion in stance phase; AMSw = maximum angle of ankle dorsiflexion in swing phase; A-ROMst = range of motion of ankle joint during stance phase. Range of motion was computed as the difference between the maximum and minimum (AmSt index) values of the plot. Foot, FP-IC = foot progression angle at initial contact; FP-MEAN = mean value of foot progression. All kinematic graphs obtained during the gait analysis were normalized as the percentage of the gait cycle, producing sagittal kinematic plots of the pelvis, hip, knee and ankle for each cycle. For such, the BTS Smart-D Clinic software (BTS, Milan, Italy) was used and the data were exported to txt. files.

The Six-Minute Walk Test (6 MWT) was performed to determine the degree of mobility and the speed to be used on the treadmill. As described elsewhere [16], this is a simple test developed to evaluate functional capacity through the measure of the distance traveled in a given period of time. Each volunteer was instructed to walk at a self-selected pace without running for six-minutes along a 30-m track. The volunteer was allowed to vary the pace and stop to rest, if necessary [17]. The test was performed twice: once for familiarization and once to record the distance travelled, which was transformed from m/s to km/h to determine the treadmill training speed. A one-week interval was respected between the first and second tests. Blood pressure and heart rate were monitored to ensure cardiovascular stability.

### 2.6. Intervention

The intervention consisted of 10 sessions at a frequency of five sessions per week over two consecutive weeks. The experimental group (EG) received active tDCS and the control group (CG) received sham stimulation administered for 20 min during treadmill training. Treadmill training was performed on the Inbramed treadmill (Millennium ATL, Rio Grande do Sul, Brazil) without body weight support. Training speed was established based on the results of the 6 MWT. The distance travelled on this test was recorded and transformed from m/s to km/h. The speed was then separated into 60% and 80% for each individual in both groups. The 6 MWT was only performed to determine the training speed and the results were not used for the statistical analysis. Treadmill training began with 60% of the average speed for each individual, was increased to 80% after five minutes and maintained until the final two minutes, when the speed was reduced to 60% again to avoid the abrupt interruption of physical effort [10]. Blood pressure and heart rate were monitored to ensure cardiovascular stability.

A transcranial stimulation device (DC-STIMULATOR, NeuroConn, Ilmenau, Germany) was used with two-sponge (non-metallic) surface electrodes measuring 5 cm^2^ moistened with saline solution. For active tDCS, the stimulation current was set at 2 mA and applied for 20 min. Intensity and duration were selected based on previous studies demonstrating these parameters to be safe [15,16,17] and effective at inducing a persistent increase in cortical excitability, as measured by motor evoked potentials in the tibialis anterior muscle during transcranial magnetic stimulation [18]. The CG received sham tDCS for 20 min with current set to ramp up and then down over the first 75 s of the session [19]. This sham protocol produces the same sensation as active tDCS without having a measurable effect on cortical excitability and is widely used to ensure subject blinding in investigations using tDCS [19]. The anodal electrode was placed over the primary motor cortex (M1) of the damaged hemisphere (C3 or C4) and the cathode was positioned in the supraorbital region on the contralateral side (Fp1 or Fp2), following the International10/20 EEG system, which is widely used as a guide in studies involving tDCS for stroke survivors [20]. Previous studies involving the C3/C4 montage report positive effects on lower limb function [10,21,22,23]. Testing different electrode montages, Manoli et al. (2017) [24] found that the C3/C4 montage is more beneficial and safer for individuals in the chronic phase following a stroke. The stimulator enables the entering of a programming code for either active or sham procedures. A member of the team who did not participate in the evaluations was in charge of entering the code and controlling the electrode montage for each individual. Another member of the team was in charge of switching on the device and monitoring the individual during treadmill training. This procedure was performed to ensure the blinding of both the participant and therapist. At the end of each training session, the researcher in charge of training recorded all occurrences for the determination of any adverse effects from either stimulation or treadmill training.

### 2.7. Sample Size

Sample size was calculated using the STATA (StatSoft Inc., Tulsa, OK, USA) [25]. Gait speed was selected as the primary outcome. The main change in gait among stroke survivors is a reduced walking speed, which is directly related to kinematic variables during the gait cycle [26]. Based on data in the literature [27] as well as mean and standard deviation values during the pre-intervention (0.47 ± 0.17 m/s) and post-intervention (0.78 ± 0.26 m/s) evaluations, 11 subjects in each group would be needed for a bi-directional alpha of 0.05 and an 80% test power, to which 20% was added to compensate for possible dropouts. Thus, the final sample was 14 individuals in each group (total: 28). All the previously defined parameters were computed for each participant in the two groups. The sample size was calculated following the model described in the literature [28].

### 2.8. Data Analysis

Intention-to-treat analysis was performed a priori. The Kolmogorov-Smirnov test was used to determine the distribution of the data. As non-normal distribution was found, median values and the interquartile range (IQR) of all indices were calculated for each group. Friedman’s repeated-measures ANOVA was used to test differences among Evaluations 1, 2, 3 and 4, as well as to determine whether a specific treatment introduced statistically significant changes (intra-group analysis). The Mann Whitney test was used to evaluate differences between groups at baseline (pre-intervention) and Evaluations 2, 3 and 4 (inter-group analysis). A *p*-value < 0.05 was considered indicative of statistical significance. The data were organized and tabulated using the IBM^®^ SPSS^®^ Statistics (version 26.0; IBM Corp., Armonk, NY, USA, Chicago, IL, USA) [29].

## 3. Results

Twenty-eight patients were randomized and analyzed (fourteen patients per group). The baseline characteristics of the participants are displayed in Table 1. All procedures of the study adhered to the CONSORT guidelines and are summarized in a flowchart (Figure 1).

The results of the spatiotemporal gait variables in the two groups are summarized in Table 2. No significant improvements (*p* > 0.05) in these variables were found among the evaluations in either group.

The results of the spatiotemporal gait variables in the two groups are summarized in Table 2. No significant improvements (*p* > 0.05) in the spatiotemporal variables were found among evaluations in either group (Figure 2).

Kinematic gait variables in the two groups are displayed in Table 3. No statistically significant differences were found on the non-paretic side in the two groups (Table 3). For better visualization of the results, Table 3 only displays kinematic variables on the paretic side. Significant improvements were found for the pelvis, hip, knee and ankle (*p* < 0.05) (Table 3). In the EG, the intra-group analysis revealed significant improvement in kinematic knee variables (KMSw and K-ROM) (Figure 3) and ankle variables (AMSt, A-ROMSt and AMSw) (Figure 4) at Evaluations 2, 3 and 4 (Table 3). The inter-group analysis revealed statistically significant differences in the EG in comparison to the CG regarding a kinematic knee variable (KMSw) at Evaluation 2, kinematic hip variable (HAA-ROM) and kinematic pelvis variable (PT-IC) at Evaluations 2 and 3 (Table 3). No statistically significant differences were found in the CG for any of the variables evaluated on either the paretic or non-paretic side (*p* > 0.05) (Table 3).

## 4. Discussion

The present original study was conducted to analyze the effects of anodal tDCS with a current of 2 mA administered over the damaged M1 with the cathode positioned in the supraorbital region on the contralateral side combined with simultaneous treadmill training on spatiotemporal and kinematic gait variables in stroke survivors.

The statistical analysis showed no significant differences in spatiotemporal variables (self-selected walking speed, step width, step length, support phase and double support phase) in either group. In contrast, statistically significant differences were found in the EG regarding kinematic gait variables of the paretic limb, specifically the angle of pelvic tilt at initial contact (PT-IC), range of motion of hip abduction/adduction (HAA-ROM), maximum angle of knee flexion in the swing phase (KMSW), range of motion of the knee (K-ROM), minimum angle of ankle plantar flexion in the stance phase (AmSt), maximum angle of ankle dorsiflexion in the swing phase (AMSw) and range of motion of the ankle joint in the stance phase (A-ROMst). Active tDCS combined with treadmill training had immediate effects. These effects were increased after 10 sessions and maintained one month after the intervention compared to baseline regarding the kinematic variables KMSw, K-ROM, AMSt, AMSw and A-ROMst. Moreover, the EG demonstrated immediate effects and after 10 sessions regarding the kinematic variables HAA-ROM and PT-IC in comparison to the CG.

Gait speed chosen as the primary outcome of this study. Although no statistically significant differences were found using the proposed protocol with regards to this variable, the significant improvements in secondary results (kinematic gait variables) are encouraging. As described in the literature [28], gait speed is a complex functional activity and a type of multimodal product of many processes. One of the hypotheses for the significant improvement in gait speed in stroke survivors is the restoration of the range of motion of the joints during the gait cycle (improvement in kinematic gait variables) [30,31]. Rehabilitation methods that demonstrate potential for the recovery of gait speed in stroke survivors involve repetitive activities, such as treadmill training [32]. This may explain the lack of positive results in the present study, in which only the effects of one and ten sessions of tDCS combined with treadmill training were analyzed.

In the acute phase of stroke, early activation occurs in the contra-lesional cortex, with the gradual return of normal ipsi-lesional activation in the sub-acute and chronic phases. During this process, disturbances occur in neural pathways, causing impaired motor function of the limbs, which alters gait dynamics [31]. Good gait performance requires adequate cognitive function, which may also be affected after a stroke, but the change in gait pattern among stroke survivors is directly related to muscle activity. Spasticity is one of the characteristics of a stroke, which mainly affects gait speed, consequently altering gait kinetics and kinematics [30,33].

According to Guzik et al. [32], insufficient dorsiflexion upon initial contact and knee flexion in the swing phase of the paretic limb are striking kinematic characteristics of stroke survivors. The mechanical objective of dorsiflexion is to position the foot correctly to facilitate the rolling process of the ankle, leading to better propulsion of the limb. The mechanical objective of knee flexion during the swing phase is to decelerate the propulsion of the limb to facilitate the next initial contact of the same limb. In the present study, the EG exhibited improvements in these aspects in the paretic limb, suggesting improvements in kinematic gait variables and, consequently, functional mobility.

Some studies have raised the hypothesis the tDCS combined with motor training can lead to a reduction in spasticity and consequent improvements in joint movements in stroke survivors [33,34]. In a recent meta-analysis, however, Elsner et al. [35] reported that there is only a low to moderate degree of evidence suggesting that tDCS can assist in the reduction of spasticity in this population.

As hemiparetic gait causes changes in the stance and swing phases triggered mainly by inadequate plantar/dorsiflexion [36], the proposed protocol had positive effects directly related to an improvement in the range of motion of the ankle. Few studies in the literature have investigated the effects of tDCS on the lower limbs of stroke survivors [10]. In a similar protocol to that employed in the present study, Grecco et al. [11] evaluated the effects of treadmill training combined with anodal tDCS over the M1 with a current of 1 mA on gait variables in children with cerebral palsy and found statistically significant improvements in gait pattern and functional mobility, which is in agreement with the present findings.

In a pilot study, Geroin et al. [37] combined anodal tDCS with a current of 1.5 mA over the primary motor cortex and robot-assisted gait training in ten 50-min sessions held over a two-week period. The authors report statistically significant results regarding functional mobility (evaluated using the 6 MWT and the Ten-Meter Walk Test) only in the intra-group comparison. However, this improvement was maintained at the follow-up evaluation one month after the end of the ten-session protocol.

Park et al. [38] submitted 24 stroke survivors to ten sessions of tDCS over the M1 with a current of 2 mA thirty minutes per day combined with motor therapy and found significant improvements in gait. In a study involving healthy individuals and stroke survivors, Kaski et al. [39] instructed the participants to walk up and down an escalator that was switched off during tDCS over M1 and found greater cortical excitability in the paretic limb of the stroke survivors. This finding is similar to the present results regarding improvements in both knee and ankle movements in the paretic limb of the participant submitted to active tDCS. In another study, the authors also found statistically significant improvements in ankle movements of stroke survivors following the use of anodal tDCS over the damaged M1 [40].

### 4.1. Summary & Contributions

The present findings demonstrate that the combination of these two therapeutic modalities had positive effects on both ankle and knee movements on the paretic side of stroke survivors during gait. These results are in line with data described in the literature. In studies involving children with cerebral palsy, the combination of tDCS and motor training has favored gait, mobility [11] and balance [41,42]. Studies involving stroke survivors report positive results regarding cortical excitability and active ankle movements on the affected side as well as improvements in functional mobility [40].

### 4.2. Limitations and Future Work

Further studies are needed to confirm the present findings. We emphasize the importance of assessing muscle activity along with three-dimensional gait analysis after tDCS combined with treadmill training, as non-sagittal movements are determinants of gait mechanics in stroke survivors [32]. This study has limitations that should be considered. First, only data related to spatiotemporal and kinematic variables were investigated, whereas no evaluations of kinetics were performed. The lack of tools for evaluating muscle activity is also a limitation. Moreover, the lack of an evaluation of the severity of the disease and cognitive alteration constitutes another limitation of this study.

## 5. Conclusions

One and ten sessions of anodal tDCS with a current of 2 mA administered over the damaged M1 with the cathode positioned in the supraorbital region on the contralateral side combined with simultaneous treadmill training did not have significant effects on spatiotemporal variables in stroke survivors. However, the same protocol promoted significant improvements in kinematic variables of the pelvis, hip, knee and ankle in this population and results were maintained one month after treatment.

## Figures and Tables

**Figure 1 brainsci-13-00011-f001:**
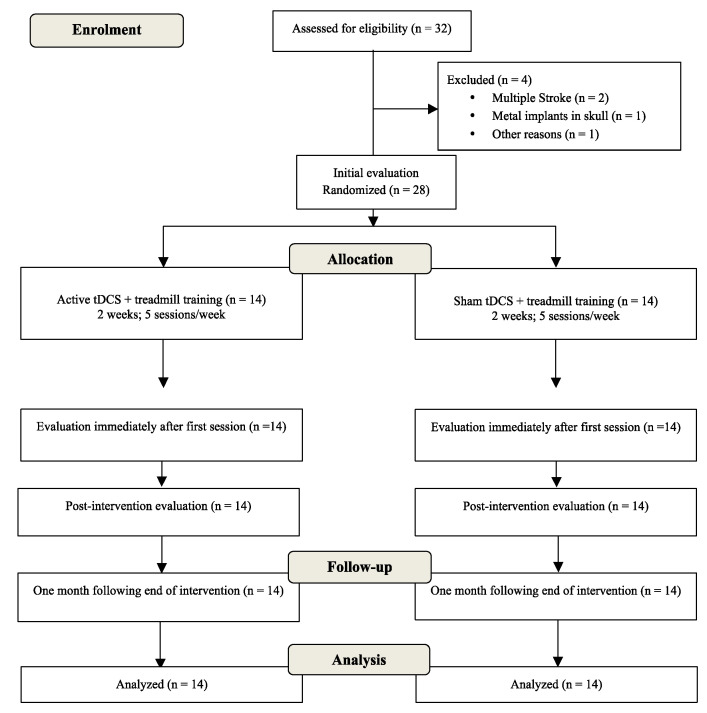
Flowchart of study based on Consolidated Standards of Reporting Trials (CONSORT).

**Figure 2 brainsci-13-00011-f002:**
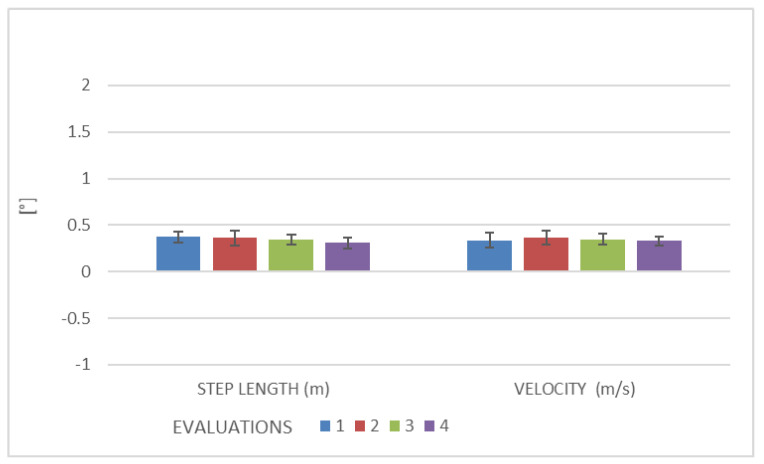
Spatio-temporal gait variables (step length and velocity) expressed as mean and standard deviation.

**Figure 3 brainsci-13-00011-f003:**
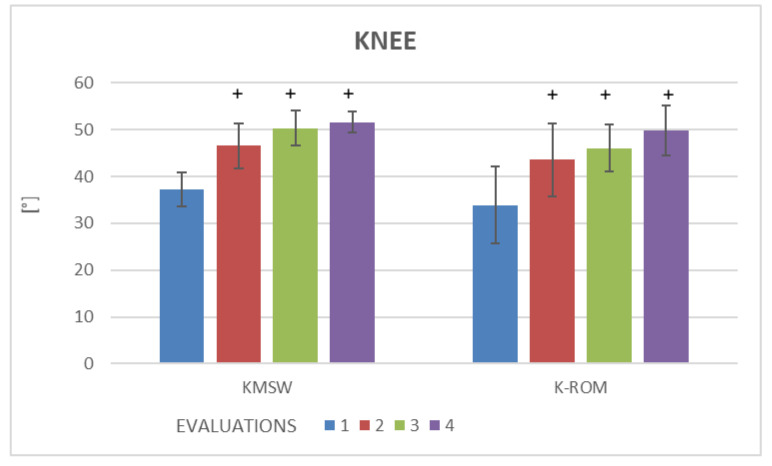
Outcomes of kinematics gait variables, for knee joint of the paretic side, expressed as median values and the interquartile range. + Statistically significant difference in comparison to baseline (*p* < 0.05). KMSW = maximum angle of knee flexion in swing; K-ROM = range of motion of knee on the sagittal plane.

**Figure 4 brainsci-13-00011-f004:**
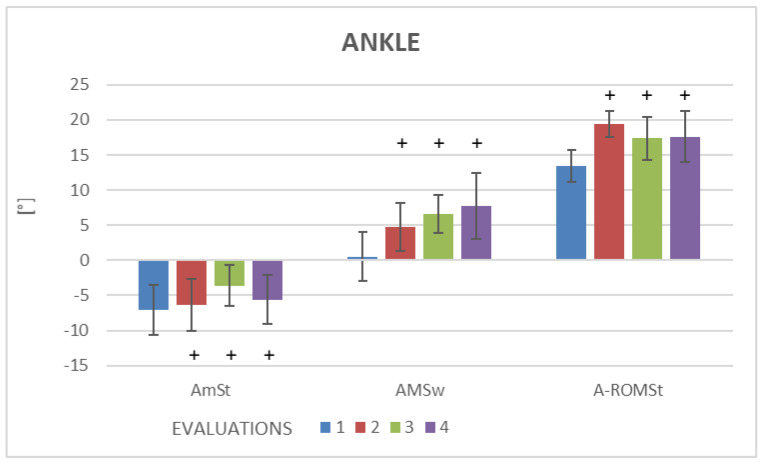
Outcomes of kinematics gait variables, for ankle joint of the paretic side, expressed as median values and the interquartile range. + Statistically significant difference in comparison to baseline (*p* < 0.05). AmMSt = minimum angle of ankle plantar flexion in stance; AMSw = maximum angle of ankle dorsiflexion in swing; A-ROMst = range of motion of ankle in stance.

**Table 1 brainsci-13-00011-t001:** Mean (standard deviation) of anthropometric characteristics in experimental and control groups.

	Groups	
	Experimental	Control
Participants (*n)*	14	14
Age (years)	58.5 (10.04)	58.4 (11.44)
Body mass (kg)	72.3(13.8)	64.9 (13.2)
Height (m)	1.69 (0.10)	1.62 (0.11)
BMI (kg/m^2^)	27.3 (6.8)	29.1 (7.4)
Time since stroke (months)	39.2 (20.2)	25.5 (19.4)
Male/female	9/3	4/8
Type of stroke (ischemic/hemorrhagic)	8/4	7/5
Affected side (right/left) Six-Minute Walk Test Gait-assistance device Cane/Braces	6/6 109.3 (37.9) 6/1	4/8 97.33 (46.6) 5/1

Legend: Mean ± SD. (*n*) = number, (kg) = kilogram, (m) = meter, (kg/m^2^) = kilogram/square meters.

**Table 2 brainsci-13-00011-t002:** Outcomes of spatiotemporal gait variables expressed as mean and standard deviation (± SD).

	Experimental Group								Control Group							
	Paretic				Non-Paretic				Paretic				Non-Paretic			
Evaluation	1	2	3	4	1	2	3	4	1	2	3	4	1	2	3	4
Stance (%Cg)	61.390 (8.938)	63.153 (8.213)	63.404 (7.543)	62.046 (7.523)	67.809 (5.989)	67.829 (3.687)	68.246 (5.813)	67.525 (4.683)	62.059 (5.943)	61.695 (4.918)	61.059 (5.839)	62.670 (3.612)	70.034 (7.101)	71.483 (6.303)	68.897 (5.232)	70.047 (5.887)
Double Support (%Cg)	17.543 (6.877)	19.149 (8.232)	16.360 (5.959)	15.283 (5.150)	12.924 (5.360)	19.721 (6.292)	16.354 (5.939)	18.634 (8.250)	15.152 (4.241)	14.391 (3.047)	14.353 (4.415)	15.204 (3.753)	14.706 (4.465)	18.799 (4.921)	16.174 (2.272)	16.962 (3.917)
STEP LENGTH (M)	0.372 (0.100)	0.363 (0.082)	0.347 (0.108)	0.307 (0.133)	0.438 (0.144)	0.400 (0.097)	0.383 (0.155)	0.371 (0.108)	0.325 (0.118)	0.369 (0.081)	0.358 (0.084)	0.354 (0.091)	0.322 (0.108)	0.316 (0.103)	0.368 (0.109)	0.383 (0.126)
Evaluation	1		2		3		4		1		2		3		4	
Velocity (M/S)	0.334 (0.167)		0.364 (0.144)		0.346 (0.113)		0.328 (0.101)		0.453 (0.288)		0.383 (0.103)		0.398 (0.114)		0.364 (0.133)	
Step Width (M)	0.414 (0.339)		0.436 (0.332)		0.422 (0.314)		0.314 (0.166)		0.200 (0.039)		0.206 (0.033)		0.311 (0.243)		0.280 (0.224)	

Legend: m = meters; m/s = meters per second; % CG = percentage of gait cycle.

**Table 3 brainsci-13-00011-t003:** Kinematics gait variables expressed as median and interquartile range.

		Experimental Group				Control Group			
		Side—Paretic				Side—Paretic			
Evaluation		1	2	3	4	1	2	3	4
**Pelvis**	**PT-IC**	13.3 (7.4)	14.7 (7.4) **#**	13.6 (5.4) #	11.3 (9.3)	9.0 (4.8)	8.4 (3.8)	9.2 (2.7)	8.6 (4.1)
	**PT-MAX**	14.425 (6.78)	14.13 (6.08)	12.901 (5.65)	13.101 (5.03)	12.72 (4.98)	13.09 (5.20)	13.0 (5.18)	11.63 (4.15)
	**PT-MIN**	6.758 (5.07)	6.093 (5.82)	5.876 (3.99)	5.56 (6.71)	7.12 (4.04)	7.58 (4.76)	7.41 (4.26)	7.15 (4.65)
	**PT-ROM**	10.9 (6.4)	10.5 (6.7)	10.6 (6.8)	9.8 (6.0)	7.8 (4.6)	8.2 (5.3)	7.8 (5.0)	7.2 (4.7)
	**PO-MAX**	6.430 (6.45)	8.70 (9.10)	8.863 (8.13)	7.51 (7.31)	5.04 (4.14)	5.174 (5.195)	2.093 (8.90)	3.233 (8.29)
	**PO-MIN**	−5.43 (3.61)	−6.83 (7.24)	−4.211 (7.32)	−5.76 (6.54)	−3.131 (6.47)	−4.66 (6.81)	−3. 595 (7.53)	−4.58 (9.11)
	**PO-ROM**	7.7(3.5)	8.5 (4.3)	7.9 (3.7)	6.4 (5.9)	6.1 (2.8)	6.6 (2.91)	6.6 (2.11)	6.2 (3.9)
	**PR-MAX**	10.73 (7.32)	12.004 (6.24)	12.002 (6.37)	11.51 (7.37)	4.414 (9.72)	4.57 (6.70)	5.64 (7.39)	5.91 (5.95)
	**PR-MIN**	−2.045 (9.24)	−0.74 (8.363)	0.209 (12.01)	−0.29 (11.14)	−6.16 (10.91)	−4.573 (9.19)	−4.60 (6.58)	−6.275 (8.05)
	**PR-ROM**	13.7 (5.5)	12.4 (5.9)	14.3 (6.2)	13.9 (6.7)	12.8 (5.8)	13.0 (6.2)	13.8 (6.5)	13.4 (6.3)
**Hip**	**HIC**	26.2 (11.7)	30.9 (13.5)	30.0 (12.7)	28.7 (7.5)	34.1 (10.1)	33.6 (10.5)	36.9 (7.2)	31.8 (11.1)
	**HMSt**	9.45 (10.65)	5.77 (21.86)	6.69 (19.85)	4.99 (19.69)	12.5 (13.5)	13.2 (17.0)	14.2 (16.6)	12.7 (17.2)
	**HmSt**	1.0 (14.5)	4.2 (10.3)	6.9 (9.0)	6.9 (5.9)	7.75 (10.16)	5.28 (8.80)	8.08 (11.32)	5.12 (12.09)
	**HMSt-ROM**	9.85 (8.79)	5.011 (18.46)	5.13 (18.26)	7.42 (18.07)	9.30 (9.25)	8.204 (7.91)	9.21 (10.80)	8.57 (11.51)
	**HAA-MAX**	7.18 (5.33)	8.462 (14.75)	8.147 (15.92)	9.10 (13.22)	5.41 (8.95)	7.06 (8.761)	8.16 (12.31)	8.833 (10.95)
	**HAA-MIN**	−0.700 (11.36)	−2.403 (8.05)	−2.043 (12.70)	−1.72 (13.10)	−3.002 (4.56)	−2.48 (10.36)	−1.732 (13.20)	−2.66 (13.02)
	**HAA-ROM**	9.7(5.5)	7.9 (4.6) #	8.9 (5.7) #	7.3 (4.0)	6.2 (3.5)	6.1 (3.0)	6.4 (3.2)	6.8 (2.9)
	**HROT-IC**	6.58 (8.46)	5.33 (11.52)	6.62 (5.90)	7.40 (9.97)	7.61 (17.06)	5.65 (19.17)	6.805 (18.0)	8.601 (23.95)
	**HROT-MEAN**	8.14 (7.30)	8.285 (15.57)	7.76 (12.07)	7.35 (16.98)	6.905 (10.28)	5.875 (20.19)	6.602 (14.25)	7.700 (11.39)
**Knee**	**KIC**	6.0 (18.45)	7.4 (14.91)	9.3 (12.67)	6.9 (9.23)	16.2 (11.47)	16.6 (12.18)	17.0 (13.45)	16.0 (12.78)
	**KMSW**	37.2 (7.22)	46.6 (9.45) +#	50.3 (7.49) +	51.6 (4.58) +	40.2 (12.02)	39.7 (10.37)	39.2 (10.53)	40.7 (10.92)
	**KMST**	3.3 (13.93)	3.1 (13.13)	4.2 (9.65)	1.8 (8.89)	5.3 (6.31)	7.0 (12.10)	5.8 (11.95)	5.3 (6.40)
	**K-ROM**	33.9 (16.45)	43.6 (17.84) +	46.1 (10.02) +	49.8 (10.79) +	34.9 (14.51)	32.8 (12.60)	33.4 (13.11)	35.4 (13.19)
**Ankle and Foot**	**AIC**	−6.0 (6.53)	−3.9 (6.141)	−2.9 (5.87)	−3.6 (6.78)	−4.5 (8.31)	−4.1 (7.19)	−2.9 (6.36)	−2.7 (9.11)
	**AMSt-MAX**	6.4 (5.92)	13.0 (8.13)	13.8 (8.21)	12.0 (7.65)	9.1 (5.74)	10.5 (15.10)	8.2 (11.34)	8.8 (11.11)
	**AMSt-MIN**	−7.1 (10.21)	−6.4 (7.31) +	−3.6 (5.88) +	−5.6 (7.02) +	−4.1 (9.25)	−1.5 (15.23)	−4.1 (10.01)	−4.7 (11.91)
	**AMSw**	0.5 (7.02)	4.7 (6.871) +	6.6 (5.40) +	7.7 (9.37) +	0.5 (7.28)	5.3 (14.24)	2.9 (8.24)	2.6 (9.11)
	**A-ROMSt**	13.4 (5.14)	19.4 (3.82) +	17.4 (6.10) +	17.6 (7.21) +	14.2 (6.21)	12.0 (7.22)	12.3 (6.13)	13.5 (7.48)
	**FP IC**	0.8 (23.12)	−0.1 (21.09)	−0.1 (24.50)	1.1 (20.52)	−6.6 (13.97)	−5.6 (11.05)	−7.3 (8.22)	−6.4 (13.86)
	**FP MEAN**	−0.5 (22.16)	−2.5 (23.54)	−2.1 (25.19)	−3.9 (18.46)	−9.8 (13.93)	−8.4 (10.28)	−7.1 (14.22)	−8.3 (11.79)

Legend: Ankle and Foot: AIC = angle of ankle dorsiflexion/plantar flexion at initial contact; AMSt = maximum angle of ankle dorsiflexion in stance phase; AmSt = minimum angle of ankle plantar flexion in stance phase; AMSw = maximum angle of ankle dorsiflexion in swing phase; A-ROMst = range of motion of ankle in stance phase; FP IC = foot progression angle at initial contact; FP MEAN = mean of foot progression. Knee: KIC = angle of knee flexion at initial contact; KMSW = maximum angle of knee flexion in swing phase; KmST = minimum angle of knee flexion in stance phase; K-ROM = range of motion of knee on sagittal plane; Hip: HIC = angle of hip flexion at initial contact; HMSt MAX = maximum angle of hip flexion/extension in stance phase; HMSt MIN = minimum angle of hip flexion/extension in stance phase; HMSt-ROM = range of motion of hip flexion/extension in stance phase; HAA-MAX = maximum angle of hip abduction/adduction; HAA-MIN = minimum angle of hip abduction/adduction; HAA-ROM = range of motion of hip abduction/adduction; HROT-IC = range of motion of hip rotation at initial contact; HROT-MEAN = mean of hip rotation; Pelvis: PT-IC = angle of pelvic tilt at initial contact; PT-MAX = maximum angle of pelvic tilt; PT-MIN = minimum angle of pelvic tilt; PT-ROM = range of motion of pelvic tilt; PO-MAX = maximum angle of pelvic obliquity; PO-MIN = minimum angle of pelvic obliquity; PO-ROM = range of motion of pelvic obliquity; PR–MAX = maximum angle of pelvic rotation; PR-MIN = minimum angle of pelvic rotation; PR-ROM = range of motion of pelvic rotation. **#** Statistically significant difference in comparison to sham stimulation (*p* < 0.05). + = *p* < 0.05, if compared to baseline.

## Data Availability

Not applicable.

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
