# Peer review of "Effects of Transcranial Direct Current Stimulation Combined with Treadmill Training on Kinematics and Spatiotemporal Gait Variables in Stroke Survivors: A Randomized, Triple-Blind, Sham-Controlled Study"

_brainsci, 2022, doi:10.3390/brainsci13010011_

Round 1

Reviewer 1 Report

This RCT investigated the effect of tDCS combined with treadmill training on kinematics and spatiotemporal gait variables in stroke patients. The authors reported no significant differences in the spatiotemporal gait variables between groups. In addition, they reported a significant difference in the kinematics variables in the experimental group. The topic is really interesting, however, I have some minor comments listed below.

The introduction is written well and provides a clear background regarding the current literature and the purpose of this study.

The methods are also written according to Consort guidelines. There is a minor comment regarding the inclusion and exclusion criteria. Do the authors use any measures to eliminate any cognitive impairments such as MMSE or MOCA? Please include that in the inclusion.

The results are clear and presented well.

In the discussion section page 11, lines 353-355 the authors mentioned that “Authors should discuss the results and how they can be interpreted from the perspective of previous studies and of the working hypotheses. The findings and their implications should be discussed in the broadest context possible. Future research directions may also be highlighted”

I think this is a typo and should be removed.

The discussion needs some improvement, for example, the authors ignore the main limitations of this RCT. They should include the main limitation and future direction and recommendations.

The conclusion seems general, the authors should make it more specific and directed depending on the results.

Author Response

Dear Editor,

The authors appreciate all the comments and suggestions made by the reviwers. It was very important to improve this file.

REVISOR 1:

This RCT investigated the effect of tDCS combined with treadmill training on kinematics and spatiotemporal gait variables in stroke patients. The authors reported no significant differences in the spatiotemporal gait variables between groups. In addition, they reported a significant difference in the kinematics variables in the experimental group. The topic is really interesting, however, I have some minor comments listed below.

Reviewer's comment: “The introduction is written well and provides a clear background regarding the current literature and the purpose of this study”.

Response: We appreciate. Thank you.

Reviewer's comment: “The methods are also written according to Consort guidelines. There is a minor comment regarding the inclusion and exclusion criteria. Do the authors use any measures to eliminate any cognitive impairments such as MMSE or MOCA? Please include that in the inclusion”.

Response: We appreciate your comment, and we agree that a specific assessment of cognitive aspects is of great importance. However, we did not use any assessment for the cognitive aspects, and we consider this as a limitation of the study.

Reviewer's comment: The results are clear and presented well.

Response: We appreciate your comment.

Reviewer's comment: “In the discussion section page 11, lines 353-355 the authors mentioned that “Authors should discuss the results and how they can be interpreted from the perspective of previous studies and of the working hypotheses. The findings and their implications should be discussed in the broadest context possible. Future research directions may also be highlighted” I think this is a typo and should be removed”.

Response: We appreciate the comment. The authors deleted the sentence of the text.

Reviewer's comment: “The discussion needs some improvement, for example, the authors ignore the main limitations of this RCT. They should include the main limitation and future direction and recommendations”.

Response: We appreciate your comment. We include the main limitations of the present study and suggestions for future studies. ‘Further studies are important to confirm these findings of the present study, we emphasize the importance to assess muscle activity, along with a three-dimensional gait assessment after the application of tDCS combined with treadmill training, once the non-sagittal movements were determinant for the gait mechanics in stroke [30]. This study has limitations that should be considered. First, only data related to spatiotemporal and kinematic variables were investigated, whereas no evaluations of kinetics were conducted, the lack of tools for evaluating muscle activity is also a limitation. Moreover, we recognize the lack of an evaluation of the severity of the disease and cognitive alteration as a limitation of the study’.

Reviewer's comment: “The conclusion seems general, the authors should make it more specific and directed depending on the results”.

Response: We appreciate your comment. We change the conclusion to: ‘The protocol of one and ten sessions of anodal tDCS with a current of 2mA administered over the injured M1 with the cathode positioned in the supraorbital region on the contralateral side combined with the simultaneous use of treadmill training did not have statistically significant effects on spatiotemporal variables in stroke patients. However, the same protocol had statistically significant effects on kinematic variables of pelvis, hip, knee and ankle not only accelerated, but improved, and results are maintained one month after treatment in this population’.

Reviewer 2 Report

Dear Authors; I found  this work an interesting assessment on  the effects of anodal transcranial direct current stimulation (tDCS) combined 24 with treadmill gait training on spatiotemporal and kinematic variables in stroke patients using gait 25 speed as the primary outcome. Majority of its writing and analysis seems solid to me. It needs some extra work prior to further  processing. Regards. P.S.

[1] Writing:

1-1  References: Make them in MDPI format. For example years are in bold font.

1-2 Abbreviations: Add list of used abbreviations in the end right before reference section.

1-3 Missing study outline paragraph: Add this at the end of Introduction section.

1-4 STAT 12 line 245: needs citation in the references. 

*StataCorp. 2021. Stata Statistical Software: Release 17. College Station, TX: StataCorp LLC.

1-5 Line 265: SPSS 19 needs citation.

*IBM Corp. Released 2020. IBM SPSS Statistics for Windows, Version 27.0. Armonk, NY: IBM Corp

1-6 Line 353-356 in the Discussion section are redundant. Remove them. 

1-7 Subsectionize the discussion section to improve its reading flow:"4.1. Summary & contributions"; "4.2. Limitations"; "4.3. Future Work".  Add the last subsection.

[2] Statistical:

2-1  Missing Barcharts:  Make a bar chart with error bars for the key messages given in Table.2 The table in current form is very detailed and hard to figure out its main message.

Example: Link: See last Figure:

https://statistics.laerd.com/spss-tutorials/bar-chart-using-spss-statistics-2.php

2-2 Same comment as in 2-1 for Table.3 main comparisons and message. 

Author Response

REVISOR 2:

Dear Authors; I found  this work an interesting assessment on  the effects of anodal transcranial direct current stimulation (tDCS) combined with treadmill gait training on spatiotemporal and kinematic variables in stroke patients using gait speed as the primary outcome. Majority of its writing and analysis seems solid to me. It needs some extra work prior to further  processing. Regards. P.S.

Reviewer's comment: “References: Make them in MDPI format. For example years are in bold font”.

Response: We appreciate your comment. We corrected the references according to the MDPI format.

Reviewer's comment: “Abbreviations: Add list of used abbreviations in the end right before reference section”.

Response: As suggested a list of used abbreviations was added in the end of the text.

Reviewer's comment: “Missing study outline paragraph: Add this at the end of Introduction section”.

Response: The authors re writen the sentence.

Reviewer's comment: “STAT 12 line 245: needs citation in the references.  *StataCorp. 2021. Stata Statistical Software: Release 17. College Station, TX: StataCorp LLC”.

Response: The reference was added as suggested.

Reviewer's comment: “Line 265: SPSS 19 needs citation. *IBM Corp. Released 2020. IBM SPSS Statistics for Windows, Version 27.0. Armonk, NY: IBM Corp”.

Response: The reference was added as sugested.

Reviewer's comment: “Line 353-356 in the Discussion section are redundant. Remove them”. 

Response: We appreciate the comment. We removed them.

Reviewer's comment: “Subsectionize the discussion section to improve its reading flow:"4.1. Summary & contributions"; "4.2. Limitations"; "4.3. Future Work".  Add the last subsection”.

Response: We appreciate your comment. We include the main limitations of the present study and suggestions for future studies at the end of the discussion. ‘Further studies are important to confirm these findings of the present study, we emphasize the importance to assess muscle activity, along with a three-dimensional gait assessment after the application of tDCS combined with treadmill training, once the non-sagittal movements were determinant for the gait mechanics in stroke [30]. This study has limitations that should be considered. First, only data related to spatiotemporal and kinematic variables were investigated, whereas no evaluations of kinetics were conducted, the lack of tools for evaluating muscle activity is also a limitation. Moreover, we recognize the lack of an evaluation of the severity of the disease and cognitive alteration as a limitation of the study’.

[2] Statistical:

Reviewer's comment: “Missing Barcharts:  Make a bar chart with error bars for the key messages given in Table.2 The table in current form is very detailed and hard to figure out its main message.

Example: Link: See last Figure: https://statistics.laerd.com/spss-tutorials/bar-chart-using-spss-statistics-2.php”.

Response: We appreciate the comment. A bar chart was included.

Reviewer's comment: “Same comment as in 2-1 for Table.3 main comparisons and message”.

Response: We appreciate the comment. A bar chart was included.

Round 2

Reviewer 2 Report

Dear Authors; my main concerns were addressed satisfactorily. Regards.